# OpenReview forum: "Evidence for Limited Metacognition in LLMs"
_ICLR.cc/2026/Conference — ICLR 2026 Poster_

### Official Review · Reviewer_yLeh · 2025-10-30

**Soundness:** 3
**Presentation:** 3
**Contribution:** 3
**Rating:** 6
**Confidence:** 3

**Summary:**

This paper investigates metacognitive abilities in LLMs, taking inspiration from research on metacognition in nonhuman animals. The authors adopt two task paradigms that measure metacognition without explicit self-reports, and find somewhat mixed results: recent models show modest success, but there is no clear, consistent evidence that the tested models are showing strong metacognitive abilities.

**Strengths:**

Overall, I liked this paper. The contribution of evaluating metacognition without self-reports is a nice addition to the literature. The writing is clear, the analyses are well-motivated and sound, and the conclusions are appropriate given the findings. I appreciated that the authors included surface features in the regression model in Section 2.4. The expected outcomes and logic laid out in Section 3.2 are also great, and lacking in most AI papers these days.

**Weaknesses:**

I like that the authors eschew self-reports, but the paper could be strengthened by providing more motivation for using implicit measures of metacognition. The current motivation in the second paragraph of the Introduction feels a bit weak. For example, I don’t understand the argument laid out by the following sentence: “Because LLMs have vast memory capacities and are trained on a nontrivial fraction of everything humans have ever written with the singular goal of generating plausible and pleasing responses, they are almost preternaturally ill-suited to trustworthy self reports” (l. 045-048). Why does having a large memory capacity, or being trained on a large amount of data, make the model ill-suited for self-reports? Since implicit measures of metacognition might introduce additional task demands, I think a bit more work needs to be done to motivate them.

**Questions:**

A few minor suggestions/questions:
- Section 2.1 (“Models”) is quite hard to parse. This information could maybe be better presented in a table, with the main text just highlighting the high-level motivation for how these models were chosen, and any theoretically relevant differences between them.
- In the Discussion, the authors write: “Speculating, as with the lack of advantage for factual knowledge in metacognition, the relatively poor performance in the self-modeling task may relate to the fact that LLMs don’t have the equivalent of the hippocampus, which in mammals subserves both the explicit recollection of facts and the ability to simulate one’s own behavior” (l. 473-476). This claim feels like a big logical jump. I doubt the authors are implicitly claiming that models need brain-like functional architectures to achieve mammals’ abilities, but this claim hints at that. Also, to my knowledge, we just don’t know (without further mechanistic studies) whether LLMs have hippocampus-like “regions” or not.
- You might want to check out relevant papers by Song et al.: “Language Models Fail to Introspect About Their Knowledge of Language” (COLM 2025), and “Privileged Self-Access Matters for Introspection in AI” (preprint 2025).

---

> ### Author Response · Authors · 2025-11-17
>
> We thank the reviewer for their perceptive comments. We have added some additional explanation for our motivations for eschewing self-reports in LLMs to the Introduction and Discussion sections. We suggest that having been exposed to such a wide expanse of possible rhetorical paths, while having the capacity to remember (a compressed version of) them, endows models with the ability to give what appears as convincingly introspective responses but are in reality responses drawn from introspective texts in its memory, pattern-matched to the input context. Our paradigms are aimed to get models “out-of-distribution” by requiring them to map their meta-cognitive knowledge, if they have it, to arbitrary and novel responses. This does entail putting additional task demands on the models, but these are well within the abilities of the frontier and near-frontier models we test, as evidenced by the facts that they all respond with outputs (eg, multiple choice options) that the tasks require, and that in the Delegate Game they all show sensitivity to teammate ability level.
>
> Regarding Section 2.1, we have converted the models list into a table, and added motivations into the text, as suggested.
>
> Regarding our comments about the mammalian hippocampus in the Discussion, while we are entirely agnostic as to architectural implementation, there is ongoing research into ways of enhancing LLMs with hippocampal-like explicit/episodic memory (eg, https://arxiv.org/pdf/2404.11672v2, https://arxiv.org/abs/2502.06975), and the recent Google paper on nested (continual) learning (https://abehrouz.github.io/files/NL.pdf) specifically mentions the hippocampus as inspiration. Our suggestion that such methods could endow LLMs with more human-like metacognitive abilities is admittedly speculative, but we feel it’s worth putting on readers’ radar, given our expressed concerns regarding the safety of self-aware AI systems.
>
> We thank the authors for alerting us to the papers from Song and colleagues. While the operationalization of introspection in the COLM paper is different from our own, we believe that the lack of introspective ability found there is consistent with our findings of weak to nonexistent introspective abilities in the older and smaller models among our overall highly capable set. We appreciate the distinction made in the preprint, and feel that the contrast between information available to the model being tested and a third party is key (and akin to Binder et al.’s (2024) approach); the finding that models cannot introspect on sampling temperature given that definition makes sense. We have added references to both in our Related Works/Discussion sections.

---

### Official Review · Reviewer_WzKm · 2025-10-31

**Soundness:** 2
**Presentation:** 3
**Contribution:** 2
**Rating:** 4
**Confidence:** 2

**Summary:**

The paper introduces two behavioral paradigms—Delegate Game and Second Chance Game—to test metacognition in LLMs without relying on self-reports. In Delegate, a model must decide to answer a question or delegate to a teammate; metacognition is inferred if delegation correlates with model-internal confidence (proxied by baseline correctness or token-probability entropy) after controlling for surface difficulty cues. In Second Chance, a model is told its previous answer was wrong and must re-answer; metacognition/self-modeling is inferred if it changes appropriately beyond neutral-prompt change rates and beyond alternative strategies (random choice, implausible-option choice, added noise). Experiments span many frontier models released since 2024 and two datasets (GPQA and SimpleQA) plus format-converted variants (GPSA and SimpleMC). The main finding: limited, context-dependent evidence for metacognitive abilities—partial correlations typically ≤0.3 in Delegate (correctness) and up to ~0.5 using entropy; Second Chance shows modest, model-specific gains after ruling out alternatives for some OpenAI models. The authors argue post-training may shape these abilities and provide a public code release.

**Strengths:**

1. Careful design and statistical controls target internal confidence rather than surface difficulty or stylistic priors.

2. (confidence-use vs. self-modeling) provide complementary views; the figure suite clearly visualizes small but significant effects.

3. Breadth of models and format-matched datasets (MC and short-answer) aid generality claims; reproducibility details and code availability help follow-on work.

**Weaknesses:**

1. Effect sizes are small and variable. Delegate partial correlations peak around 0.3 (correctness) / ~0.5 (entropy), and elicitation is inconsistent across formats and models; some provider-specific RLHF “personalities” (over-answering) confound interpretation.

2. Short-answer scoring uses LLM-as-judge. Although cross-provider panels are used, human adjudication or stricter exact-match normalization would strengthen claims.

3. Tasks are QA-centric and relatively short context; it’s unclear whether effects hold for planning-heavy, long-horizon, or tool-use settings.

**Questions:**

1. How sensitive are Delegate/Second-Chance effects to minor wording or teammate-history changes?

2. Can you harmonize decoding across models (e.g., deterministic MC with calibrated proxies) to reduce comparability confounds?

3. Could you run the same paradigms with humans (and report comparable partial correlations/change-lifts) to contextualize effect sizes?

4. Do findings persist on chain-of-thought-heavy reasoning or multi-step tool-use tasks where internal planning signals might be stronger?

---

> ### Author Response · Authors · 2025-11-17
>
> We thank the reviewer for their thoughtful comments. Regarding the smallness and variability of the effect sizes, we quite agree, and have taken pains to note that throughout (including alluding to it in our title). That result is in fact consistent with what has been found in other recent work using different paradigms, e.g., Lindsay’s[1] finding that introspective ability is “highly unreliable and context-dependent”, and Binder et al.’s[2] report of modest boosts to performance by introspection-finetuning models, and only on simple tasks within the training domains. Thus, convergent evidence suggests that that’s just where LLM’s introspective abilities are at this point. The apparent non-introspective RLHF “personality”-driven responses are not a confound in our paradigm, as the paradigm is designed to separate out the usage of internal signals from such non-introspective factors, but rather one specific factor that we suggest models rely on as an alternative to introspection. Our analysis in Appendix 10 indicates that the Delegate Game is better at eliciting reliance on internal signals than simpler paradigms.
>
> Regarding use of LLM-as-judge for the short-answer questions, the three-judge LLM panel (all frontier models from different providers) was unanimous 97% of the time on the factual SimpleQA dataset, and human review of a sample of those never found a case where the majority or unanimous choice was clearly wrong. Unanimity was less frequent on the short-answer reasoning (GPSA) dataset, at 88%; we expect that the frontier models we use are in fact better judges than humans at this extremely challenging dataset - certainly their scores are higher than we could hope to achieve. Regardless, mistakes on the part of the panel only bear on calibration, not introspection; the same panel and procedure were used in both the baseline test and game conditions.
>
> Regarding the QA, short-context nature of our tasks, this was an intentional choice. While it would indeed be interesting to investigate introspection in more extended tasks, this would add considerably to experimental complexity. It is also important in our approach that the introspection be done entirely in a single forward pass; if the model is outputting many tokens before deciding what to do, as would likely be the required for more complex tasks, it becomes difficult to disentangle the degree to which decisions are due to a model’s assessment of its internal states rather than inferences it draws from its outputs.
>
> TBC...

---

> ### Author Response · Authors · 2025-11-17
>
> Regarding sensitivity to teammate-history changes in the Delegate Game and prompt wording in the Second Chance Game: we’ve done a follow-up analysis to assess this, rerunning MC/factual and short-answer/reasoning datasets on weakly and strongly introspective models using a variety of random seeds. In our setup, this will change both the teammate history and the actual questions asked (as both are drawn mutually exclusively from the same set). The results, shown in the new Appendix 11, are within the confidence intervals established by resampling presented in the paper, and reproduce the basic pattern. We did not optimize the simple Second Chance game prompt at all, but we did run the paradigm with a less exhortative version (no system prompt), and find modestly lower change rates but the same qualitative pattern. (We also did not optimize prompt wording in the Delegate Game, but qualitatively we observed that wording changes could impact delegation frequency, but not reliance on internal signals.)
>
> Regarding the decoding, we are not entirely certain what is meant by “calibrated proxies”. To clarify our own procedure for the baseline capabilities test: for all models tested, the highest probability answer was chosen for the partial correlation analysis shown in Figure 1. For large models that did not return logprobs and were sampled at temperature 0 and for models that did return logprobs and were sampled at temperature 1, these were exactly equivalent, as temperature does not affect the ordering of responses but only their absolute probability. For smaller models that did not return logprobs and for which we used resampling at temperature 1 to estimate probabilities these were approximations; as we resampled until the CIs of the estimates were <=5 percentage points wide, the approximations should be very close to reality. During the Delegate Game, all models were sampled at temperature 0 and during the Second Chance Game (where entropy is important for our subsequent analysis) we followed the same logic as for the baseline tests; the text in the Methods section has been updated to note this.
>
> We agree that comparable experiments with humans would be enlightening and help us assess how far along LLMs are on the metacognitive spectrum, and we do indeed intend this as future work. However we feel it is beyond the scope of this paper. Such experiments have not been conducted in other investigations into LLM introspection in the literature, and are not necessary to establish the existence of such abilities in LLMs, which is our primary purpose here (in addition to introducing a new empirical framework). We’d like to inspire more work in the field in that direction though, which is why we call it out in the Discussion.
>
> [1] https://transformer-circuits.pub/2025/introspection/index.html
> [2] https://arxiv.org/pdf/2410.13787

---

### Official Review · Reviewer_WeDC · 2025-10-31

**Soundness:** 3
**Presentation:** 4
**Contribution:** 3
**Rating:** 6
**Confidence:** 3

**Summary:**

This paper explores whether there is evidence that frontier LLMs have metacognitive abilities largely through behavioral evaluations. More specifically, the authors investigate self-modeling using two behavioral paradigms: the delegate game and the second chance game. They define self-modeling as the ability to monitor and control one's internal states and propose that their evaluations inspired by non-human animal studies are more reliable than self-reports. Their results suggest limited evidence of graded, metacognitive abilities in LLMs, which the authors suggest are qualitatively different from human metacognition.

**Strengths:**

The paper is original in its proposal of a novel non-linguistic methodology from studies in non-human animals for LLMs to evaluate self-modeling abilities. In particular, the fact that the method bypasses the need for self-reports from the LLMs makes it a significant contribution in the right direction for designing evaluations of this kind. Although the results are suggestive at best, the methodology is clear and the analyses sound.

**Weaknesses:**

The paper's central claim is finding "evidence for limited metacognition", but the supporting evidence is consistently described by the authors themselves as "limited in resolution". Given the modest effect, it remains highly plausible that other subtle, non-introspective strategies (e.g., learned heuristics that associate "your answer was incorrect" with "pick the next most likely token") could be responsible for the small performance lift. The paper does not provide a strong affirmative case for why self-modeling is a more parsimonious explanation than some unknown, simpler heuristic. Additionally, the authors don't address what seems like a fundamental question about whether it even makes sense to ask this question about self-modeling or awareness given the architecture that LLMs run on (i.e., lack of recursion, feedback connections, and so on). Finally, some more work on situating the behavioral results using mechanistic interpretability would solidify the findings if further support was found there.

**Questions:**

- What might be some other heuristics that you have considered might explain the behavior of the modest effect presented in the paper but are difficult to evaluate, and why?
- What does it mean for an LLM to have a "partial" self-modeling ability? Your results and discussion highlight the differences with human metacognition, but it's not clear what it means if a model is not acting fully in accordance with what you would expect from an agent that has metacognitive abilities.
- What kinds of mechanistic interventions would provide stronger evidence that the models are indeed demonstrating metacognitive abilities?

---

> ### Author Response · Authors · 2025-11-17
>
> We thank the reviewer for their thoughtful comments. And we appreciate the skepticism towards LLMs’ introspective abilities. In fact we were motivated to initiate this research in part by the same skepticism, and the sense that overly credulous interpretations of self-reports were leading some to impute to LLMs abilities that they did not have. However, having designed paradigms to eliminate as many confounds as we could and still seeing a consistent pattern of above-chance performance across models, we have had to update our priors. No experiment is perfect, but we feel this work does advance the field towards more objective evaluations, as you suggest, and provides valuable evidence that real metacognitive abilities are emerging. The finding that these abilities are “limited” and “partial” is consistent with what has been found in other recent work using different paradigms, e.g., Lindsay’s[1] finding that introspective ability is “highly unreliable and context-dependent”, and Binder et al.’s[2] report of modest boosts to performance by introspection-finetuning models, and only on simple tasks within the training domains. Convergent evidence suggests that that’s just where LLM’s introspective abilities are at this point. We think it’s important to emphasize both the evidence and its limitedness.
>
> Part of our initial skepticism was also driven by the architectural limitations of transformers, as you suggest, but having gotten some pushback in solicited feedback on an earlier draft of this paper, we removed mention of those so as to avoid getting derailed on arguments that were not crucial to our paper.
>
> Regarding alternative explanations, this is an important consideration. We did endeavor to test these. We came up with three different alternative explanations regarding performance in the Second Chance Game, and test them all in the paper. It is conceivable that the models have been trained to “pick the next most likely token” when told they are wrong, but the ability to identify on the fly which specific token that is in a given context seems to us to be a noteworthy introspective ability. A difficult-to-test-for potential alternative explanation for performance in the Delegate Game is that models have been post-trained to know what kind of questions they are good or bad at, and rely on that stored knowledge rather than introspection during the game. However, this would seem more applicable to the reasoning (GPQ/SA) questions than the random obscure factual SimpleQA/MC ones, and performance is just as high on those. (We also have another paradigm we are developing in ongoing work that might obviate this concern.) Another potential confound we’ve thought of is competition between answer tokens mechanically biasing the delegation decision; however we test an alternative version of the paradigm that asks only for a decision token and find that models can also succeed at it, albeit at a lower rate. We thank the reviewer for bringing this up, and have added a discussion on this theme to a new “Limitations and future research” section.
>
> We agree that white box analysis would complement the black box analysis performed here. A challenge with that is that the most notable examples of introspection come from proprietary and/or very large models. One potential way around that is by fine-tuning introspective ability into smaller, open-weight models, an avenue we are currently exploring in separate work. Whether in small, fine-tuned or large models, we will seek to identify internal correlates of metacognitive representations, and to establish causality by manipulating them and measuring how that affects tasks performance.
>
> [1] https://transformer-circuits.pub/2025/introspection/index.html
> [2] https://arxiv.org/pdf/2410.13787

---

### Author Response · Authors · 2025-12-01

We thank all of the reviewers for their comments, and regret that circumstances have ended the conversation prematurely. Our summary follows.

All three reviewers acknowledged the originality of our methodological contribution - adapting non-linguistic paradigms from animal cognition research to evaluate metacognition in LLMs without relying on self-reports - and the robustness of our analysis. Two reviewers note the modesty and variability of the effects we report. We agree with this assessment, and believe that both aspects are important facts about current LLMs' metacognitive abilities. We note that others, using quite different paradigms (e.g., Lindsey, 2025; Binder et al., 2024), also report modest and variable introspective abilities in LLMs, which provides convergent evidence for this picture. We also add a number of controls for confounds, to err on the side of conservatism.

To directly address specific reviewer comments, we have clarified wording regarding decoding procedures in the Methods section, converted the model list into a table, added motivation for implicit measures to the Introduction, and expanded the Discussion with a new "Limitations and future research" section discussing heuristics we considered and why introspection remains a parsimonious account of our findings. We have also conducted additional analysis on prompt and teammate-history sensitivity, reported in a new Appendix 11, which confirms that our results in both the Delegate Game and Second Chance Game remain robust across different random seeds and wording variations. Finally, we report in a new Appendix 12 results from a variation of the Delegate Game that rules out one proposed alternative non-introspective explanation, and find that the qualitative pattern of results remains. We believe that the clarified wording, expanded discussion, and additional analysis have strengthened our argument and enhanced our paper, and we appreciate the reviewers’ feedback on these points.

---

### Meta-Review · Area_Chair_9FWg · 2026-01-06

**Summary:**

This paper introduces two behavioral paradigms (Delegate Game and Second Chance Game) to evaluate metacognitive-like behavior in LLMs without relying on self-reports. Across a broad set of recent frontier models, the authors find modest, context-dependent signals consistent with limited self-monitoring / self-modeling, and complement the behavioral evidence with token-probability analyses. Reviewers broadly agreed the methodology is original and the experiments are generally careful and well-controlled.

**Reviewer Concerns:**

1. The evidence for “metacognition” is modest and could be explained by alternative heuristics, so the interpretation may be overstated (Reviewer WeDC, WzKm, yLeh).
2. Results may be sensitive to prompt wording, teammate history, and cross-model comparability/decoding choices, raising robustness concerns (Reviewer WzKm).
3. The study’s scope is QA/short-context-centric, with requests for human baselines and/or mechanistic evidence to strengthen claims (Reviewer WzKm, WeDC).
4. The paper needs clearer motivation for implicit behavioral measures vs. self-reports, and some presentation/discussion elements felt weak or speculative (Reviewer yLeh).

**Reviewer Scores:**

I think all the reviewers will maintain their initial scores.

---

### Decision · Program_Chairs · 2026-01-26

Accept (Poster)